# Assessment of Inter-Institutional Post-Operative Hypoparathyroidism Status Using a Common Data Model

**DOI:** 10.3390/jcm10194454

**Published:** 2021-09-28

**Authors:** Joon-Hyop Lee, Suhyun Kim, Kwangsoo Kim, Young Jun Chai, Hyeong Won Yu, Su-Jin Kim, June Young Choi, Yoo Seung Chung, Kyu Eun Lee, Ka Hee Yi

**Affiliations:** 1Gil Medical Center, Department of Surgery, Gachon University College of Medicine, Incheon 21565, Korea; deftnovice@gmail.com (J.-H.L.); dryooseung@hanmail.net (Y.S.C.); 2Department of Applied Statistics, Chung-Ang University, Seoul 06974, Korea; suhy0118@gmail.com; 3Transdisciplinary Department of Medicine & Advanced Technology, Seoul National University Hospital, Seoul 03080, Korea; 4Department of Surgery, Seoul Metropolitan Government Seoul National University Boramae Medical Center, Seoul 07061, Korea; 5Department of Surgery, Seoul National University Bundang Hospital, Seongnam 13620, Korea; hyeongwonyu@gmail.com (H.W.Y.); aznagran@gmail.com (J.Y.C.); 6Department of Surgery, Seoul National University College of Medicine, Seoul National University Hospital, Seoul 03080, Korea; su.jin.kim.md@gmail.com (S.-J.K.); kyueunlee@snu.ac.kr (K.E.L.); 7Department of Internal Medicine, Seoul Metropolitan Government Seoul National University Boramae Medical Center, Seoul 07061, Korea; imykh@naver.com

**Keywords:** common data model, hypoparathyroidism, incidence, risk factors, thyroidectomy

## Abstract

Post-thyroidectomy hypoparathyroidism may result in various transient or permanent symptoms, ranging from tingling sensation to severe breathing difficulties. Its incidence varies among surgeons and institutions, making it difficult to determine its actual incidence and associated factors. This study attempted to estimate the incidence of post-operative hypoparathyroidism in patients at two tertiary institutions that share a common data model, the Observational Health Data Sciences and Informatics. This study used the Common Data Model to extract explicitly specified encoding and relationships among concepts using standardized vocabularies. The EDI-codes of various thyroid disorders and thyroid operations were extracted from two separate tertiary hospitals between January 2013 and December 2018. Patients were grouped into no evidence of/transient/permanent hypoparathyroidism groups to analyze the likelihood of hypoparathyroidism occurrence related to operation types and diagnosis. Of the 4848 eligible patients at the two institutions who underwent thyroidectomy, 1370 (28.26%) experienced transient hypoparathyroidism and 251 (5.18%) experienced persistent hypoparathyroidism. Univariate logistic regression analysis predicted that, relative to total bilateral thyroidectomy, radical tumor resection was associated with a 48% greater likelihood of transient hypoparathyroidism and a 102% greater likelihood of persistent hypoparathyroidism. Moreover, multivariate logistic analysis found that radical tumor resection was associated with a 50% greater likelihood of transient hypoparathyroidism and a 97% greater likelihood of persistent hypoparathyroidism than total bilateral thyroidectomy. These findings, by integrating and analyzing two databases, suggest that this analysis could be expanded to include other large databases that share the same Observational Health Data Sciences and Informatics protocol.

## 1. Introduction

Post-thyroidectomy hypoparathyroidism is an unintended sequela that may result in transient or permanent symptoms ranging from tingling sensation to severe breathing difficulties if left untreated [1]. The incidence of iatrogenic hypoparathyroidism has been reported to range from 7% to 37% [2,3], a range too broad to be indicative. This variability in incidence stems not only from inter-operator variability, but also from differences in thyroid operations performed in various medical communities.

The pattern of thyroid surgery in South Korea differs slightly from that in other parts of the world. Because medical services in South Korea are readily accessible to the general population, thyroid cancer tends to be detected at an early stage and its incidence is very high, about 51.1 per 100,000 persons in 2017 [4]. Moreover, about 68.5% of operations for thyroid cancer in 2019 were performed by a high volume surgeons at tertiary medical centers due to the high density and urban orientation of the population of South Korea [5].

The concentration of thyroid patients to large tertiary centers suggests that a thorough analysis of their data can provide meaningful and generalizable results about the incidence of and factors associated with thyroid cancer and post-thyroidectomy hypoparathyroidism. Hospital-specific clinical data warehouses (CDWs), which were developed to efficiently support data analysis, are platforms used to integrate complex data sources through specialized analytical tools [6,7,8]. These in-hospital CDWs, which facilitate data processing and analysis, can be used for financial, administrative, clinical, and research purposes, including for hypothesis generation and retrospective analysis. However, synthesizing CDW data from multiple tertiary centers requires a common data model (CDM).

The purpose of this study was to determine the incidence of post-thyroidectomy hypoparathyroidism at two major tertiary hospitals in South Korea sharing a common CDM. Furthermore, this study utilized the CDM data to determine the pre-operative factors associated with iatrogenic hypoparathyroidism.

## 2. Materials and Methods

### 2.1. Study Population

Patients who underwent total thyroidectomy at Seoul National University Hospital or Seoul National University Bundang Hospital between January 2013 and December 2018 were enrolled in this study. Patients with preoperative indications of hypoparathyroidism, defined as PTH < 10 pg/mL, were excluded. The study protocol was approved by the Institutional Review Boards of Seoul National University Hospital and Seoul National University Bundang Hospital, which waived the requirement for informed consent because this study involved minimum risk to the participants (IRB no.: E-2004-039-1116 and X-2005/615-902). Our study was performed in accordance with the Declaration of Helsinki in its latest form.

### 2.2. Data Collection

This study used the CDM adopted by the OHDSI network, which specifies how to encode and collect clinical data in detail, in multiple organizations, including explicitly specified encoding and relationships among concepts using standardized vocabularies.

Because the participants of this study had to be identified from the information provided by two separate databases, the exact operational definitions of the groups had to be clearly predefined. The control group was operationally defined as those patients who did not experience post-operative hypoparathyroidism or those who only experienced it transiently; patients with serum PTH concentrations ≥ 10 pg/mL on all assays performed within 6 months after thyroidectomy [9] or patients with serum PTH < 10 pg/mL at least once within 6 months after thyroidectomy but with PTH ≥ 10 pg/mL afterwards. Persistent hypoparathyroidism patients was defined as a serum PTH concentration < 10 pg/mL at least once within the first 6 months after thyroidectomy and at least once again 6 to 18 months after thyroidectomy.

Patients with Graves’ disease and other benign conditions were identified based on the EDI-codes, which were entered for billing and reimbursement purposes. Patients with thyroid cancer were designated using a code of C73 (thyroid cancer—the subtypes of thyroid cancers, such as papillary or follicular cancers, are not specified in the coding system) more than twice postoperatively, including the operation date. Patients with Graves’ disease were designated using a code of E050 (Graves’ disease) or E059 (hyperthyroidism) more than once and without any entry of C73 (thyroid cancer) after the thyroidectomy. Patients with other benign thyroid diseases were designated using codes of D34 (benign neoplasm), E040 (simple goiter), E041 (thyroid nodule), E042 (simple goiter, multinodular), E049 (adenomatous goiter), E051 (toxic nodular goiter), and E052 (toxic multinodular goiter).

Operation types included radical tumor resection (total thyroidectomy with lymph node dissection due to thyroid carcinoma), total bilateral thyroidectomy (total thyroidectomy only due to benign disease including Graves’ disease), total unilateral thyroidectomy (unilateral thyroid lobectomy regardless of disease type), and subtotal unilateral thyroidectomy (subtotal thyroidectomy regardless of disease).

### 2.3. Statistical Analysis

The relationships among the three groups of patients were analyzed by Pearson’s Chi-squared test. The effects of age were analyzed by Student’s t-test and analysis of variance (ANOVA). Multivariable logistic regression analyses were performed to test the associations among the three groups. All analyses were performed using the statistical program R version 3.6.0 from the Free Software Foundation, Inc. (http://cran.r-project.org/, access date: 8 January 2021), with *p*-values < 0.05 considered statistically significant.

## 3. Results

During the study period, 7616 patients underwent thyroidectomy; of these, 2768 patients were excluded, including 2619 patients who underwent unilateral thyroid lobectomy and 149 with missing post-operative parathyroid hormone (PTH) levels (Figure 1). The 4848 eligible patients, who received operations that included total thyroidectomy, consisted of 3633 (74.94%) women and 1215 (25.06%) men, with a mean age of 49.7 ± 10.2 years at operation. Among the 4848 eligible patients, 4548 (93.81%) underwent radical tumor resection and 300 (6.19%) underwent total bilateral thyroidectomy. All 4548 (93.81%) patients with thyroid cancer underwent radical tumor resection, whereas all 300 patients (6.19%) with benign conditions, including 140 (2.89%) with Graves’ disease and 160 (3.30%) with other benign conditions, underwent total bilateral thyroidectomy (Table 1). Of the total 4848 patients who underwent radical tumor resection or total bilateral thyroidectomy, 1370 (28.26%) experienced symptoms of transient hypoparathyroidism, 251 (5.18%) experienced persistent hypoparathyroidism, and 3227 (66.56%) showed no evidence of hypoparathyroidism.

Male sex was significantly more frequent in patients with persistent hypoparathyroidism (29.9%) than in patients with transient hypoparathyroidism (23.3%) (*p* = 0.025), but not more frequent than in patients without hypoparathyroidism (25.4%) (*p* = 0.121) (Figure 2). There was no difference in sex between the no hypoparathyroidism group and transient hypoparathyroidism group (*p* = 0.121). Furthermore, the proportion of patients who underwent radical tumor resection did not differ in patients with persistent (96.4%) and transient (95.2%) hypoparathyroidism, but it was significantly lower in patients without hypoparathyroidism (93.0%) than in those with both transient (*p* = 0.006) and persistent (*p* = 0.039) hypoparathyroidism (Figure 3). Among the patients who underwent total bilateral thyroidectomy, there were no statistically significant differences in the proportions with Graves’ and other benign conditions among the normal, transient, and persistent hypoparathyroidism groups.

Because diagnosis generally overlapped with operation type, diagnosis was not included in subsequent analyses to prevent collinearity. Univariate logistic analysis predicted that, relative to total bilateral thyroidectomy, radical tumor resection was associated with a 48% greater likelihood of transient hypoparathyroidism and a 102% greater likelihood of persistent hypoparathyroidism (Table 2). Moreover, multivariate logistic regression analysis found that radical tumor resection was associated with a 50% greater likelihood of transient hypoparathyroidism and a 97% greater likelihood of persistent hypoparathyroidism than total bilateral thyroidectomy (Table 3).

## 4. Discussion

The present study evaluated the incidence of hypoparathyroidism among 4848 eligible patients who underwent thyroid surgery at two tertiary centers. Among these patients, 1370 (28.26%) experienced transient and 251 (5.18%) experienced hypoparathyroidism during the study period. Because these two centers utilized a common CDM protocol, the results from these two centers could be analyzed together. Univariate analysis identified meaningful baseline factors associated with hypoparathyroidism, whereas multivariate analysis failed to identify statistically significant factors.

Our CDM protocol was based on the Observational Health Data Sciences and Informatics (OHDSI) database, which was designed for utilization in large scale observational studies [10,11]. These CDMs were developed to manage large amounts of data in the medical field, including claims and clinical data. The advantages of using a standardized CDM include the compatibility of data among different institutions, allowing the use of standard analytical tools [12,13]. In South Korea, efforts have begun to share EMR data in the form of CDMs among more than 40 tertiary medical centers throughout the country. Since 2018, these institutions have converted their EMR data to CDM format using OHDSI open-source resources [14].

In accordance with this trend, CDMs have been used for medical research in several areas, including an overview of traumatic brain injury models [15], mitochondrion-related diseases based on neurologic examinations [16], and evaluation of hospitalization and mortality rates of patients with atrial fibrillation [17]. A recent CDM-based study from South Korea on patients with inflammatory bowel disease found that prognosis was less favorable in patients with early onset than late onset disease [14]. CDM based studies related to thyroid operations or their sequelae have not yet been published.

A nation-wide insurance database study among South Korean patients who underwent total thyroidectomy in 2012 indicated that the prevalence of persistent post-operative hypoparathyroidism was 10.4%, comparable to our results [18]. Differences between studies were likely due to differences in the operational definitions of persistent hypoparathyroidism, the study periods, and the institutions. Persistent hypoparathyroidism in our study was defined as a serum PTH concentration < 10 pg/mL at least once within the first 6 months after thyroidectomy and at least once more 6 to 18 months after thyroidectomy, whereas the other study, which was based on claims data, had a more complicated operational definition based on active vitamin D prescription. A CDM analysis based on serum PTH level could objectively adjust the operational definition of hypoparathyroidism, whereas studies based on insurance claims do not provide such flexibility and rely on subjective determinations. For example, a prescription for vitamin D and calcium prescription may have resulted from concurrent osteoporosis, a situation that cannot be detected by the operational definition based on claim data. We were able to exclude such patients because we could check the serum PTH level of patients at both institutions. Finally, our analysis included patients who underwent surgery by expert high-volume thyroid surgeons, whereas the national insurance claim database contains results in patients who underwent surgery by both expert and novice surgeons, which may result in a higher incidence of hypoparathyroidism.

This study had several limitations. First, because of the retrospective nature of this cross sectional multi-institutional study, the results may have been influenced by selection or indication biases. This is more pronounced by the inclusion only of patients who underwent surgery at tertiary medical centers with high volume surgeons. Furthermore, only the parameters that were coded in accordance with the CDM protocol could be analyzed. For example, the pathology reports are not the same in the two institutions; therefore, only factors common to both could be incorporated into the database. Another example would be whether parathyroid autotransplantation was done, of which the surgical record would vary among operators. This would lead to a loss of data and may explain why our multivariate analysis did not yield any significant results. Lastly, serum vitamin D level was not included in evaluating hypoparathyroidism. Because low vitamin D level increases serum PTH level, and some patients with low vitamin D level might be excluded from the hypoparathyroidism even though they had actual hypoparathyroidism. Unfortunately, however, both institutions do not routinely measure pre-operative vitamin D level, and therefore, it was impossible to analyze its correlation to post-surgical hypothyroidism.

In conclusion, our CDM study revealed that the incidences of persistent and transient hypoparathyroidism were 5.18% and 28.3%, respectively, among patients at the two tertiary centers. Furthermore, when compared with total bilateral thyroidectomy, patients who underwent radical tumor resection were 50% and 97% more likely to experience transient and persistent hypoparathyroidism, respectively. This study not only estimated the incidence of and risk factors for post-operative hypoparathyroidism at two institutions, but also suggested that this analysis could be expanded to include other large databases that share the same Observational Health Data Sciences and Informatics protocol by integrating and analyzing the databases at the two institutions. These findings also suggested that, by integrating large institutional databases without relying on subjective factors, CDMs can produce large data results in endocrine surgery. Further CDM studies involving multiple tertiary hospitals may provide more representative and comprehensive analyses in endocrine surgery.

## Figures and Tables

**Figure 1 jcm-10-04454-f001:**
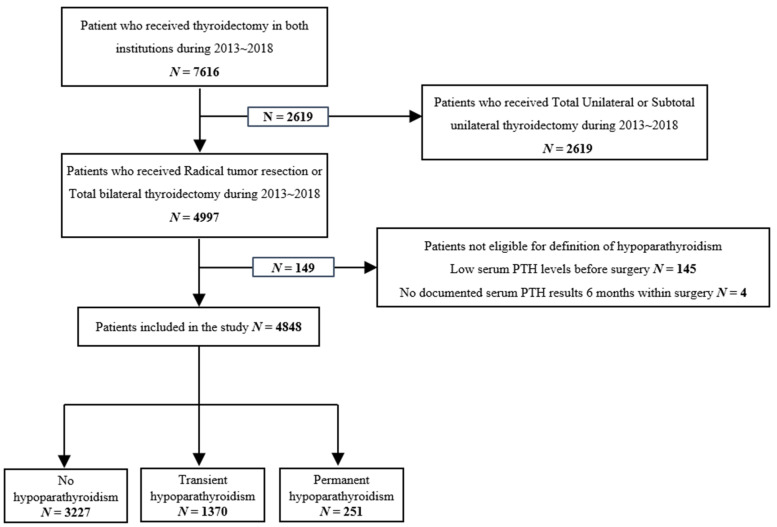
Flowchart of patient inclusion.

**Figure 2 jcm-10-04454-f002:**
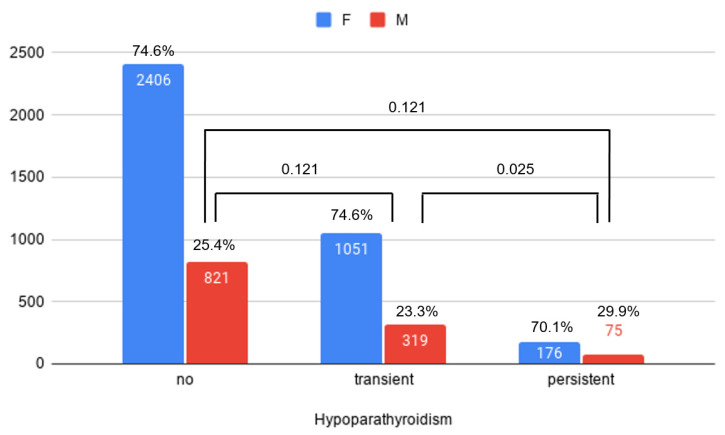
Post-operative hypoparathyroidism status according to sex. F: Female; M: male.

**Figure 3 jcm-10-04454-f003:**
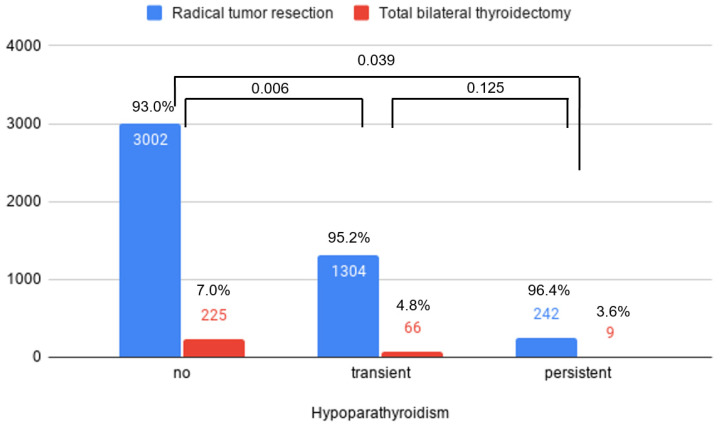
Post-operative hypoparathyroidism status according to operation type.

**Table 1 jcm-10-04454-t001:** Baseline Characteristics.

Factors		*N* (%)
Age, years		49.7 ± 10.2
Gender		
	Female	3633 (74.94)
	Male	1215 (25.06)
Operation		
	Radical tumor resection	4548 (93.81)
	Total bilateral thyroidectomy	300 (6.19)
Hypoparathyroidism		
	No	3227 (66.56)
	Transient	1370 (28.26)
	Persistent	251 (5.18)
Diagnosis		
	Thyroid cancer	3045 (93.18)
	Graves	108 (3.30)
	Other benign	115 (3.52)

**Table 2 jcm-10-04454-t002:** Univariate Logistic Analysis of Risk Factors for Transient and Persistent Hypoparathyroidism Compared to the No Hypoparathyroidism Group.

Compared to No Hypoparathyroidism Group	Risk Factors	Odds Ratio	95% CI
Transient	Radical tumor resection (vs. Total bilateral thyroidectomy)	1.48	1.12–1.96
	Male (vs. Female)	0.89	0.77–1.03
Persistent	Radical tumor resection (vs. Total bilateral thyroidectomy)	2.02	1.02–3.97
	Male (vs. Female)	1.25	0.94–1.66

**Table 3 jcm-10-04454-t003:** Multivariate Logistic Analysis of Risk Factors for Transient and Persistent Hypoparathyroidism Compared to the No Hypoparathyroidism Group.

Compared to No Hypoparathyroidism Group	Risk Factors	Odds Ratio	95% CI
Transient	Radical tumor resection (vs. Total bilateral thyroidectomy)	1.5	1.13–1.99
	Male (vs. Female)	0.88	0.76–1.02
Persistent	Radical tumor resection (vs. Total bilateral thyroidectomy)	1.97	1.00–3.89
	Male (vs. Female)	1.23	0.92–1.62

## Data Availability

Data sharing not applicable.

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
