# Peer review of "Assessment of Inter-Institutional Post-Operative Hypoparathyroidism Status Using a Common Data Model"

_jcm, 2021, doi:10.3390/jcm10194454_

Round 1

Reviewer 1 Report

  • The captions of figures 2 and 3 are in the wrong place.
  • Since you Did you included en bloc ressections? 
  • How do you explain that male sex was significantly more frequent in patients with persistent hyperparathyroidism?

Author Response

Reviewer 1

  1. The captions of figures 2 and 3 are in the wrong place.

Response 1: Thank you for comment. Captions were revised.

  1. Since you Did you included en bloc ressections? 

Response 2: En bloc resections were included in ‘radical tumor resection’

  1. How do you explain that male sex was significantly more frequent in patients with persistent hyperparathyroidism?

Response 3: I appreciate your insightful comment. Persistent hypoparathyroidism was more prevalent in male than transient group. Authors believe such statistical difference is due to the large number of study population, which made small difference in number statistically significant. In fact, the difference in absolute percent does not seem great (23.3% vs 29.9%). On the other hand, comparison between no hypoparathyroidism versus persistent group is clinically more valuable than comparison between transient versus persistent hypoparathyroidism. There was no difference in sex between no hypoparathyroidism group and persistent hypoparathyroidism group as shown in Table 2 and Table 3. Therefore, we did not emphasize male prevalence in persistent group compared to transient group in discussion.

Reviewer 2 Report

I appreciated the manuscript that I found well presented and clear. 

I.ve just a comment on the definition of postsurgical hypoparathyroidism identified and chosen by the Authors as PTH<10 pg/ml without any comment on calcium levels .. please could they add some comment on this in term of specificity and predictive value of PTH compared to calcium measurements?

Have the Authors the possibility to collect data from the OHDSI network on presurgical vitamin D status and correlate it with postsurgical hypoparathyroidism?   

Author Response

Reviewer 2

I appreciated the manuscript that I found well presented and clear. 

  1. I’ve just a comment on the definition of postsurgical hypoparathyroidism identified and chosen by the Authors as PTH<10 pg/ml without any comment on calcium levels .. please could they add some comment on this in term of specificity and predictive value of PTH compared to calcium measurements?

Response 1: Thank you for the comment. First, calcium level is not a reliable factor to diagnose hypoparathyroidism because calcium level is affected by other factors. Patients can take oral calcium for other reasons than hypoparathyroidism such as osteoporosis. Moreover, oral calcium is an over-the-counter drug, and patients can take oral calcium without prescription. For these reasons, we did not take into consideration including calcium level as an indicator of hypoparathyroidism. Second, the normal range of PTH of the two participating institutions is 15 to 65 pg/ml. In order to set a sensitive standard, we set PTH level < 10 pg/ml as a definition of hypoparathyroidism.

  1. Have the Authors the possibility to collect data from the OHDSI network on presurgical vitamin D status and correlate it with postsurgical hypoparathyroidism?   

Response 2: Thank you for the recommendation. It would be valuable to evaluate vitamin D level as an associated factor with hypoparathyroidism because vitamin D level affects PTH level. Low vitamin D level increases serum PTH level, and some patients with low vitamin D level might be excluded from the hypoparathyroidism even though they had actual hypoparathyroidism. Unfortunately, however, both institutions do not routinely measure pre-operative vitamin D level, and therefore, it was impossible to analyze its correlation to post-surgical hypothyroidism.

We have added this consideration in Discussion as follows.

Because low vitamin D level increases serum PTH level, and some patients with low vitamin D level might be excluded from the hypoparathyroidism even though they had actual hypoparathyroidism. Unfortunately, however, both institutions do not routinely measure pre-operative vitamin D level, and therefore, it was impossible to analyze its correlation to post-surgical hypothyroidism.
